# Study of the epidemiological behavior of malaria in the Darien Region, Panama. 2015–2017

**Lorenzo Cáceres Carrera**[1]*, **Carlos Victoria**[2], **Jose L. Ramirez**[3], **Carmela Jackman**[4], **José E. Calzada**[5], **Rolando Torres**[1]

**1** Department of Medical Entomology, Gorgas Memorial Institute of Health Studies, Panama City, Panama, **2** Malaria Program, Ministry of Health, Panama City, Panama, **3** Crop Bioprotection Research Unit, National Center for Agricultural Utilization Research, Agricultural Research Service, United States Department of Agriculture, Peoria, Illinois, United States of America, **4** Epidemiology Department of the Darién Region, Ministry of Health, Panama City, Panama, **5** Direcction of Research and Technological Development, Gorgas Memorial Institute of Health Studies, Panama City, Panama

* lcaceres@gorgas.gob.pa

## Abstract

### Background

Malaria is endemic in Darién and an assessment of the different factors affecting its epidemiology is crucial for the development of adequate strategies of surveillance, prevention, and disease control. The objective of this study was to determine the main characteristics of the epidemiological behavior of malaria in the Darien region.

### Methods

This research was comprised of a retrospective analysis to determine the incidence and malaria distribution in the Darien region from 2015 to 2017. We evaluated malaria indicators, disease distribution, incidence (by age group and sex), diagnostic methods, treatment, and control measures. In addition, we examined the cross-border migration activity and its possible contribution to the maintenance and distribution of malaria.

### Results

During the period of 2015–2017, we examined 41,141 thick blood smear samples, out of which 501 tested positive for malaria. *Plasmodium vivax* was responsible for 92.2% of those infections. Males comprised 62.7% of the total diagnosed cases. Meanwhile, a similar percentage, 62.7%, of the total cases were registered in economically active ages. The more frequent symptoms included fever (99.4%) and chills (97.4%), with 53.1% of cases registering between 2,000 and 6,000 parasites/µl of blood. The annual parasitic incidence (API) average was 3.0/1,000 inhabitants, while the slide positivity rate (SPR) was 1.2% and the annual blood examination rate (ABER) 22.5%. In Darién there is a constant internal and cross-border migration movement between Panama and Colombia. Malaria control measures consisted of the active and passive search of suspected cases and of the application of vector control measures.

**Data Availability Statement:** All data are contained in the manuscript.

**Funding:** LCC was assigned and administered the funds assigned by the Ministry of Economy and

Finance (MEF). No grant was received for the authors. www.mef.gob.pa. The MEF did not play any role in the design of the study, collection and analysis of data and the decision to publish or prepare the manuscript.

**Competing interests:** The authors have declared that no competing interests exist. LC is member of the Sistema Nacional de Investigación (SNI), SENACYT, Panama. Any opinions, findings, conclusions or recommendations expressed in this publication are those of the author(s) and do not necessarily reflect the view of the U.S. Department of Agriculture. The mention of firm names or trade products does not imply that they are endorsed or recommended by the U.S. Department of Agriculture over other firms or similar products not mentioned; the USDA is an equal opportunity employer.

**Abbreviations:** WHO, World Health Organization; NMP, National Malaria Program; ED, Epidemiology Department; MINSA, Ministerio de Salud; ICGES, Instituto Conmemorativo Gorgas de Estudios de la Salud; SC, Suspected case; CC, Confirmed case; AC, Autochthonous case; IMC, Imported Case; INC, Introduced case; RC, Recurrent case; RDTs, Rapid diagnostic tests; API, annual parasitic incidence; SPR, slide positivity rate; ABER, annual blood examination rate; HLC, Human landing catches; LLINs, Long lasting insecticidal nets; IRS, Indoor residual spraying; RDTs, rapid diagnostic tests; ACTs, Artemisinin-based combination therapies.

## Conclusion

This study provides an additional perspective on malaria epidemiology in Darién. Additional efforts are required to intensify malaria surveillance and to achieve an effective control, eventually moving closer to the objective of malaria elimination. At the same time, there is a need for more eco-epidemiological, entomological and migratory studies to determine how these factors contribute to the patterns of maintenance and dissemination of malaria.

## Introduction

Malaria remains one of the major tropical health challenges in the world today, with an estimated 1,382 billion people at risk of infection. Although environmental factors provide the primary conditions for the breeding of malaria vectors, the socio-demographic environment also plays a significant role in the prevalence of the disease [1,2]. This disease gravely affects the health and work capacity of a vast number of people given its wide geographic distribution, morbidity, mortality and socio-economic impact [3]. In 2017 there were 91 countries with autochthonous malaria, and an estimated 219 million malaria cases resulting in 435,000 deaths. Most of the malaria cases and 92% of the fatalities took place in Africa [4]. Malaria transmission in the Americas currently occurs in 21 countries and territories with 132 million people at risk of infection and 21 million living in high transmission risk areas. According to the World Health Organization (WHO), there were an estimated 20 million fewer malaria cases in 2017 than in 2010. Nevertheless, data for the period of 2015–2017 indicates that no significant progress was made in reducing global malaria cases in this time frame [5]. In the Americas it has been estimated that between 2000 and 2014 there was a 67% decrease in the number of malaria cases and a 79% reduction in the number of deaths [6]. Mesoamerica (It extends from approximately central Mexico through Belize, Guatemala, El Salvador, Honduras, Nicaragua, and northern Costa Rica) also experienced a decrease in malaria cases, with *P. vivax* as the predominant species and with only a small remaining foci of autochthonous (a defined and circumscribed locality situated in a currently or formerly malarious area and containing the continuous or intermittent epidemiological factors necessary for malaria transmission) *Plasmodium falciparum* transmission in the area. In contrast to the rest of the world, in particular South America, both parasite species are still susceptible to chloroquine [7]. This makes it the drug of choice for malaria treatment in most Mesoamerican countries, except Panama, where *P. falciparum* treatment is based on Arthemeter-Lumefantrine [8].

Several studies have indicated that diverse factors are associated with the incidence of malaria in endemic areas. These factors include migration, the architectural layout of a house, mosquito breeding sites near homes, human activities that increase the exposure to vectors, as well as socio-economic, cultural, demographic and epidemiological characteristics of the endemic zones [9]. It is possible that malaria outbreaks will occur in areas with frequent malaria cases but they can also occur in areas that have been declared malaria free [10]. Furthermore, the endemic regions usually present their own transmission dynamics which have to be addressed independently; considering not only the eco-epidemiological aspects but also the socio-economic conditions and specific characteristics of the population. The inclusion of these aspects increases the likelihood of success in the selection, implementation and evaluation of strategies designed to prevent and control the disease [11]. In addition, the incidence and dissemination of malaria is associated with the anthropogenic climate change, changes in land use, and control of vectors and diseases [12–14]. Changes in climate patterns leads to

changes in transmission dynamics. We can assume that an increase in malaria incidence is the result of temperature, humidity, moderate precipitation [which increases the vector life expectancy], in addition to the biological and social human-derived factors [15]. This is the reason why malaria has been considered as a disease prone to disturbances derived from global climate change. [16].

Malaria continues to represent an important public health problem in Panama due to its endemic/epidemic transmission patterns, its incidence and great socioeconomic impact in the population. Studies conducted by the National Malaria Program (NMP) have shown that the different endemic regions present distinct epidemiological conditions. During the last 40 years, the main endemic regions are concentrated in two foci near the border, from where the disease is further disseminating to other areas [17]. In Panama, the active malaria transmission is limited to 10 municipalities (around 12.5% of all Panamanian municipalities), with the rural, poor regions and mainly indigenous communities, being the most affected by this disease. The indigenous regions, which occupy 22.0% of the Panamanian territory and is house to 12.0% of the total population of the country, is where more than 85% of the total cases diagnosed at the country level are registered [18]. The variations that occur in these regions determine the main changes in the epidemiology of malaria in Panama.

Darién region, located in eastern Panama bordering Colombia, has been historically considered a malaria-endemic region, with predominant *P. vivax* infections and lately with periodical cases of *P. falciparum*, mainly from imported cases [17,19]. Despite the disease control efforts by the NMP, from the Ministry of Health (known as MINSA in spanish) during the last few years, malaria continues to be a public health problem in this region. A previous study indicated that Darién region, which includes the indigenous comarcas (The term comarca refers to an administrative region within Panama and it is assigned to a given indigenous population) Embera–Wounaan, Wargandi and Ngäbe-Bugle, is the main foci where the transmission of malaria has increased [20]. Malaria in this region keeps expanding to nearby areas, especially those that are receptive and vulnerable to malaria transmission. During the period of 2006–2017, Darién region alone registered a total of 2,211 malaria cases, which represent 23.0% of the total cases registered for Panama and an average of 184 malaria cases per year. *Plasmodium vivax* infections accounted for 95.0% of the diagnosed malaria cases in Darién. Malaria transmission in this region is very similar to what occurs in the rest of Panama, but it also displays its own characteristics where the disease is transmitted during outbreaks or foci, with variations that are strongly associated to the migrant movement, climate, ecology, socio-cultural patterns, and socio-economical activities [17]. Based on the malaria occurrence during the last 12 years in Panama, we can classify it as a country with low malaria endemicity and hence with low frequency of protective immunity in its population. This also means that there is a risk for severe disease and for the development of complications at all ages, including adults. This is especially true for infections caused by *P. falciparum*. Several mosquito vector species inhabit this region, with six species thought to be associated with malaria transmission: *An. albimanus*, *An. darlingi*, *An. oswaldoi*, *An. punctimacula*, *An. pseudopunctipennis* y *An. triannulatus* [21,22].

The main socio-economic activity of Darién region is based mainly on extensive land use, especially agriculture, cattle raising, and logging. The population is comprised primarily by mestizos (a racial-cultural mix between Europeans and native Americans) [23], afro-descendants (descendants from African slaves brought to the American continent) [24], and indigenous people from the Embera-Wounaan and Guna. This region also shares a border with Colombia, which has an ineffective control of symptomatic and asymptomatic migrants. Other important factors are the presence of different ethnicities with their own cultural beliefs that hamper health care and especially the treatment of malaria.

An understanding of the temporo-spatial dimension of malaria epidemiology is essential to develop strategies to control and eliminate the disease. The objective of this study was to determine the main characteristics of the epidemiological behavior of malaria in the Darien region.

## Material and methods

### Study design

We conducted a retrospective analysis to determine the incidence and malaria distribution in the Darien region; considering all diagnosed cases via passive and active searches conducted by the NMP during the period of 2015 to 2017. From this data we estimated the number of cases, frequency, percentages, rates, measures of central tendency and data distribution.

### Description of the area studied

Darién is located in eastern Panama, bordering with Colombia (Fig 1). The malaria foci located in the eastern border with Colombia, which includes the indigenous comarcas of Guna

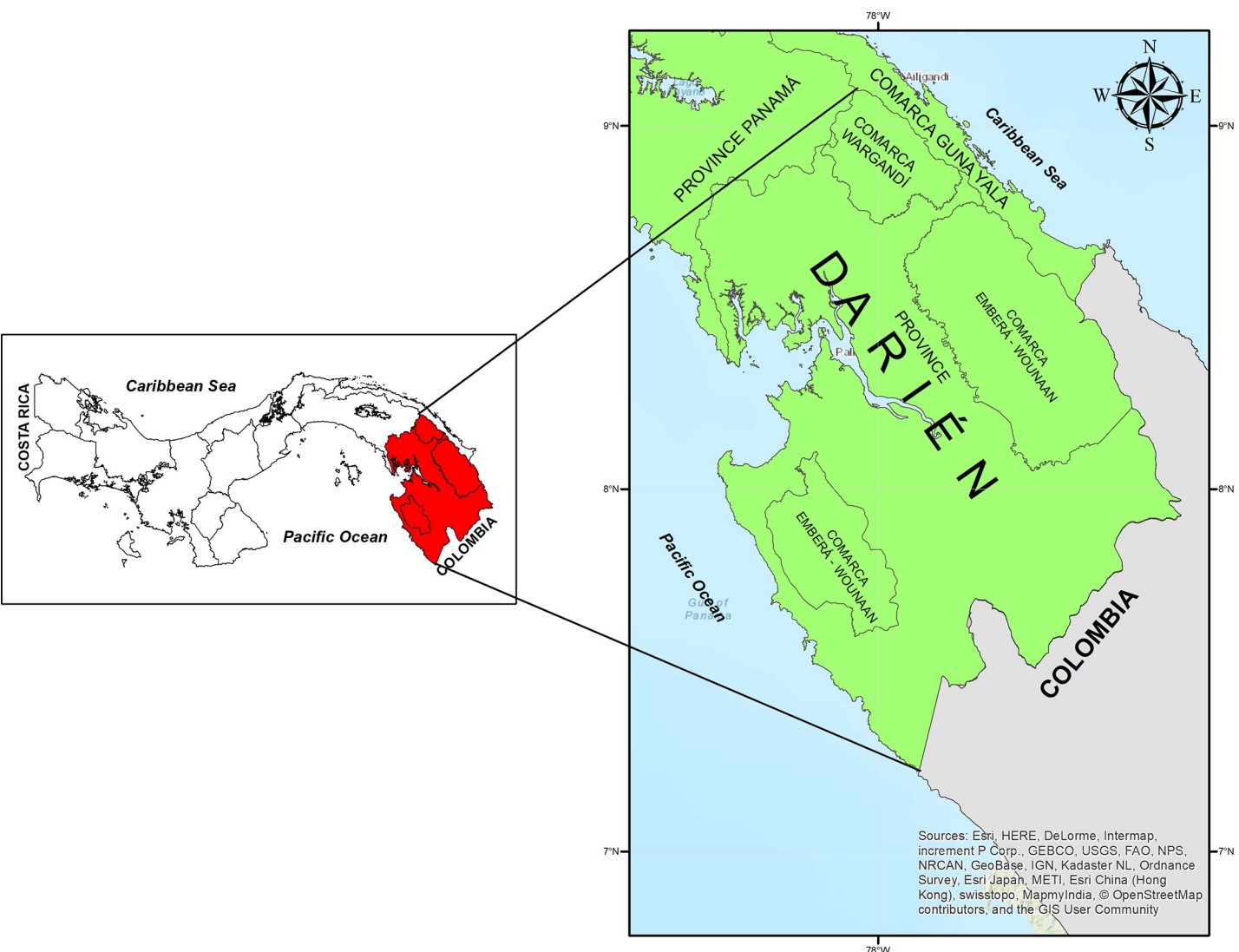

**Fig 1. Geographic location of the Darién Region, Panama.**

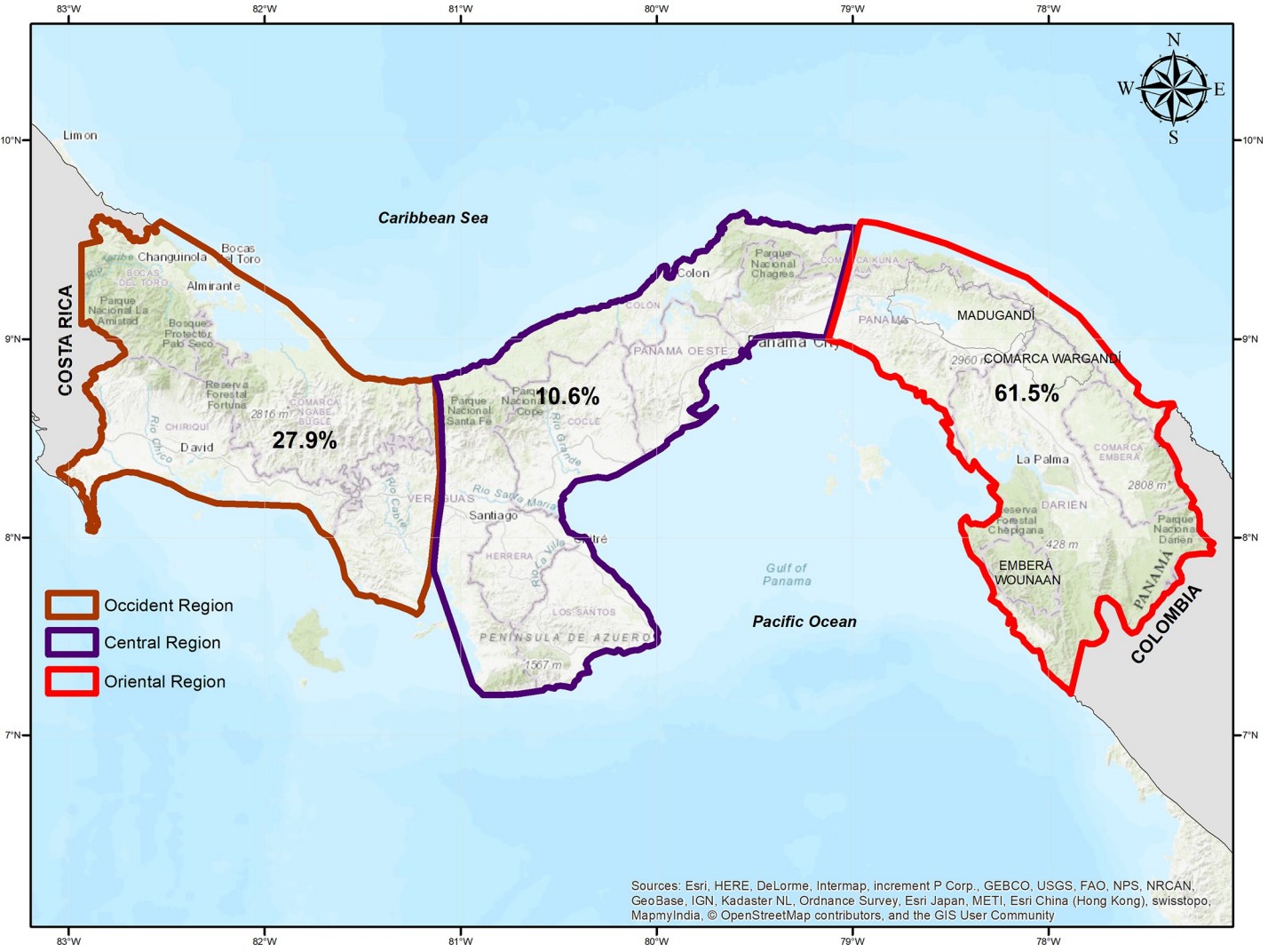

**Fig 2. Distribution map of the total malaria cases (as a percentage and by geographic region) in Panama from 2006 to 2017.**

Yala, Madungandi, Wargandi, Embera-Wounaan and Darién province, registered 61.5% of the malaria cases diagnosed in Panama from 2006 to 20017. In turn, the foci on the western border with Costa Rica, which includes the Bocas del Toro, Chiriquí provinces and the comarca Ngäbe Buglé, registered 27.9% of the total malaria cases. Lastly, the central region registered 10.6% of the total cases. This region receives imported malaria cases from the eastern and western regions due to the migration of infected persons and is linked to economic, socio and cultural activities. These cases are routinely identified by the surveillance system of this region (Fig 2). Darién climate is tropical, with characteristic high humidity, like the rest of Panama, and it is influenced by the intertropical convergence zone [25]. The region has two well-defined seasons with moderate high temperature and humidity: the dry season (January to April) and the rainy season (May to December). The average annual precipitation ranges from 1,700 to 2,000 mm, with May and November being the rainiest months. The temperature does not present significant variations, ranging from 25.6 to 27.1°C. This region registers a relative humidity of 84.0% and it ranges from 75.0% (March, dry season) to 93.0% (November,

rainy season) [26]. According to the system of biosphere classifications by Holdridge, Darién belongs to a tropical rainforest [25].

## Target population

The study was based on the incidence of malaria in Darién region between 2015 and 2017 and derived from the number of positive cases reported by the NMP. The retrospective study included all the malaria symptomatic cases positively diagnosed via thick blood smear by NMP (active search) and by health services (passive search). The NMP provided the basic information in relation to the geographical location of the malaria cases, clinical characteristics of the disease, date of the first cases, affected and exposed population, socioeconomic information, basic sanitation services, health services characteristics, demographic information and the prevention and control interventions carried out in the region.

## Diagnosis and treatment

Malaria is a required notifiable disease in Panama [27]. Each case was confirmed via parasitological diagnosis via thick blood smear or Giemsa stained peripheral blood smear [28]. All the samples were observed under a light microscope with a 1000x magnification. The morphological characteristics of the different *Plasmodium* life cycle stages were used as criteria to differentiate *P. vivax* and *P. falciparum* infections and to confirm malaria cases [29, 30]. Along with diagnostics, we considered the number of samples with gametocyte presence for further analysis given its epidemiological importance.

Diagnostics were also conducted through the use of rapid diagnostic tests (RDTs). In the positive samples, parasitemia was estimated per μl of blood. Once the infection was diagnosed as being *P. vivax*, the patient was immediately provided with a complete antimalarial treatment based on 1,500 mg of Chloroquine (CQ) and 210 mg of Primaquine (PQ). The *P. falciparum* cases were treated with Coartem® (20mg of Artemether + 120 Lumefrantine) for three days, according to the regulations and following the NMP treatment regime [27,31]. According to NMP regulations, a second blood sample is taken at 15 days post-diagnosis to confirm that the parasitemia was cleared. Furthermore, based on the epidemiological information of all the malaria-positive patients, we conducted an analysis of the main signs and clinical symptoms most frequently suffered during the disease.

## Signs and symptoms

Based on the information gathered from the epidemiological research forms from each of the reported malaria cases by *P. vivax* and *P. falciparum* respectively, we were able to learn the main malaria signs and symptoms from each of the patients.

## Categorization and definition of malaria cases

In the epidemiological characterization, malaria cases were classified according to the definitions of the Manual of Rules and Procedures for Malaria from the NMP and the epidemiological classification of malaria by WHO. Suspected case (**SC**): a person that presents a current (1–3 days) or recent (4–30 days) febrile episode, with periods of intermittent chills and sweating, who is a resident or coming from a malaria endemic area. Confirmed case (**C**): a malaria case [or malaria infection] in which the parasite has been detected via a diagnostic test [microscopy], rapid or molecular diagnostic test. Autochthonous case (**AC**): a locally-contracted malaria case, without evidence of being imported or a transmission directly originating from an imported case. Imported Case (**IMC**): a malaria case in which the infection was contracted

outside of the area of diagnostic surveillance. Introduced case (**INC**): an autochthonous malaria case with solid epidemiological tests that link it directly to a known imported case. Recurrent case (**RC**): recurrent manifestation of malaria infection [in terms of clinical symptoms and /or parasitemia] that is separated from previous manifestations of the same infection by an interval greater than any interval resulting from the normal periodicity of the malaria paroxysms [27,32,33]. The imported cases from other endemic areas of Panama that we could not distinguish were grouped within the AC. In this study, information on the number of registered malaria cases between 2015 and 2017 was provided by the NMP and the Epidemiology Department (ED) from MINSA. These cases were classified according to origin (AC, RC, IMC or INC), parasitological diagnosis (*P. vivax*, *P. falciparum* and *P. falciparum* + *P. vivax*) and according to the distribution by age and sex. The number of inhabitants and dwellings of the communities were obtained from NMP and from the National Institute of Census and Statistics respectively [34].

## Malariometric indicators

We assessed several malaria indicators such as the annual parasitic incidence (API): total number of positive slides for parasite in a year x 1000/Total population, slide positivity rate (SPR): defined as the number of laboratory-confirmed malaria cases per 100 suspected cases examined, provides an alternative method for estimating temporal changes in malaria incidence and annual blood examination rate (ABER): number of blood slides read over population at risk x 100 by year, as measures of epidemiological surveillance of malaria [35]. From the results obtained from the API, the risk level was classified at four levels: **a)**. high risk (API above 10 cases/1000 inhabitants); **b)**. medium or moderate risk (API between 1–10 cases/1000 inhabitants); **c)**. low risk (API lower than 1 case/1000 inhabitants); and **d)**. no risk (API equal to or lower than zero).

## Migration and malaria

This study briefly assessed the migratory movement and its role in the dissemination of malaria and those of antimalarial-drug resistant parasites. However, this study does not intent to identify the patterns and behavioral types of migration, nor does it quantify the magnitude of migratory movements associated with the maintenance of malaria transmission in this region.

## Climate and malaria

We considered the malaria cases diagnosed during the dry and wet season to evaluate the malaria distribution and its impact in the Darien region.

## Assessment of control measures

We contacted Darién NMP to obtain information on the control measures used in antimalarial interventions, as well as its frequency of use and coverage to control or reduce the transmission frequency.

## Statistical analysis

The data was processed with Microsoft Excel and analyzed with Tableau statistical package, version 6.0. Univariate analyses were performed for all variables. The significance difference between the mean age groups were conducted via one-way ANOVA with Tukey's multiple

comparison test. Significance was assessed at P < 0.05. This test was conducted using Prism 8, GraphPad.

### Ethical statement

This study was approved by the Committee Technical of Institutional Review the NMP of the MINSA. The search of suspected cases, the thick blood smear, the diagnosis, treatment and documentation of all the cases (detected via active and passive search), were conducted by the NMP and local health centers' technical personnel. This was done after each participant received detailed information on the procedures and after receiving verbal consent from each participant. In the case of minors, we obtained consent from the parents or legal guardians.

Epidemiological information was obtained from the NMP and ED databases. Malaria represents a major public health problem in Panama [36,37] and the results from this study will contribute to the improvement of disease surveillance and to the elimination of malaria in Panama.

## Results

### Malaria incidence

Darién has an estimated population of 55,055 inhabitants, most of whom are at risk of malaria infection, including the indigenous comarcas of Embera-Wounaan and Wargandi [34]. There are approximately 610 communities where the NMP epidemiological surveillances are conducted. From 2015 to 2017, there were a total of 23 communities with active malaria transmission. From this group a total of 41,141 thick blood smears were examine with 501 samples confirmed positive by microscopy. This represents 23.0% of the total diagnosed cases in Panama during the same period. A total of 39 *P. falciparum* cases were detected, from which 17 cases were imported from Colombia. *P. vivax* constituted 92.2% (n = 462) of the cases, where 7.8% (n = 39) corresponded to *P. falciparum* infections. Malaria cases were distributed in 23 administrative corregimientos (corregimiento refers to an administrative division of a territory in Panama) with the greatest number of cases from 2015 to 2017 occurring in comarca Wargandi (n = 102; 20.4%), Lajas Blancas (n = 86; 17.2%), Manuel Ortega (n = 83; 16.6%) and Yaviza (n = 59; 11.8%) (Tables 1 and 2). The communities with the greatest number of malaria cases in 2015 were Meteti in the corregimiento of Meteti (n = 15; 8%), Lajas Blancas (n = 13; 7.6%), and Marraganti (n = 11; 6.5%), both belonging to the administrative district of Lajas Blancas. In 2016, the communities with the greatest number of malaria cases were: Morti, in the corregimiento of Wargandi (n = 22; 12%), followed by Cocalito in the corregimiento of Jaque (n = 17; 9.2%), and Boca de Tigre in the corregimiento of Manuel Orterga (n = 13; 7.0%). In 2017, the communities of Union Choco in the corregimiento of Cirilo Guaynora (n = 17; 11.6%), Wala (n = 17; 11.6%) and Morti (n = 12; 8.2%), both in the comarca of Wargandi, were the communities with the greatest number of malaria cases. The collected data indicates an increase in the number of cases from the chronological distribution of cumulative malaria cases from 2015 to 2017 was observed in June and July and from December to March (Fig 3).

### Malariometrics indicators

The malaria indicators in the Darién region from 2015 to 2017 were: API of 3.1/1000 inhabitants; 3.4/1000 inhabitants and 2.6/1000 inhabitants respectively. The SPR was 1.0%, 1.4%, and 1.3% and the ABER was 30.5%, 24.7%, and 19.6% respectively. Table 3 shows the

**Table 1. Distribution of cases malaria by corregimientos and community in Darien Region. 2015 to 2017.**

| Province Comarca | Corregimiento | Community | 2015 | | 2016 | | 2017 | |
|---|---|---|---|---|---|---|---|---|
| | | | *P. vivax* | *P. falciparum* | *P. vivax* | *P. falciparum* | *P. vivax* | *P. falciparum* |
| Darién | Agua Fría | Agua Fría | 2 | | 2 | | | |
| | | Clarita Abajo | 3 | | | | | |
| | | Agua Fría 2 | | | 3 | | | |
| | Boca de Cupe | Boca de Cupe | 4 | | | | 1 | |
| | Río Sábalo | Boca de Sábalo | 1 | | 1 | | | |
| | Chepigana | Chepigana | | | 3 | | | |
| | Cucunati | Cucunati | 1 | | | | | |
| | El Real | Uruceco | 4 | | | | 1 | |
| | Jaque | Jaque | 1 | | | 5 | | |
| | | El Guayabo | | | | 1 | | |
| | | Cocalito | 3 | | | 16 | | |
| | | Payita | | | | 1 | | |
| | Jingurudo | Boca de Guinea | 1 | | | | | |
| | La Palma | La Palma | 2 | | 1 | | 2 | |
| | | Boca Lucas | | | 2 | | | |
| | Paya | Paya | | | | | 1 | |
| | Pinogana | Nuevo Progreso | | | 1 | | 1 | |
| | Pucuro | Pucuro | 1 | | | | 2 | |
| | | Loma Larga | | | | | 1 | |
| | Puerto Piña | Puerto Piña | | | | 3 | | |
| | Río Congo Arriba | Río Venao | | | | | 3 | |
| | Río Iglesias | Coredo | | | 1 | | | |
| | Meteti | Brazo de Sansón | 1 | | | | | |
| | | Canclón | | | 2 | | | |
| | | Canclón Arriba | | | 3 | | | |
| | | El Bongo | | | 1 | | | |
| | | Villa Darién | 1 | | | | | |
| | | Meteti | 12 | | 3 | | 3 | |
| | | Punuloso Arriba | 1 | | | | | |
| | | San Vicente | | | 1 | | | |
| | Santa Fe | Arreti | 1 | | | | | |
| | | La Villa | 2 | | | | | |
| | | Los Monos | 2 | | | | | |
| | | Santa Fe | 4 | | 1 | | | |
| | | Zapallal | | | 1 | | | |
| | Yaviza | Bella Vista | 2 | | 4 | | 5 | |
| | | Boca De Tigre | | | 2 | | | |
| | | Corozal | | | 1 | | | |
| | | Pueblo Nuevo | 10 | | 1 | | | |
| | | Quebrada Grande | 1 | | 1 | | | |
| | | Tupiza | | | 1 | | | |
| | | Yaviza | 8 | | 13 | | 14 | |
| | Yape | Yape | | | 1 | | 2 | |

malariaometric indices from 2012 to 2017. Fig 4 shows the malaria incidence distribution per corregimiento according to the API.

**Table 2. Distribution of cases malaria by corregimientos and community in Región Darien. 2015 to 2017.**

| Province Comarca | Corregimiento | Community | 2015 | | 2016 | | 2017 | |
|---|---|---|---|---|---|---|---|---|
| | | | *P. vivax* | *P. falciparum* | *P. vivax* | *P. falciparum* | *P. vivax* | *P. falciparum* |
| Embera-Wounaan | Lajas Blancas | Barranquillita | | | 2 | | | |
| | | Bajo Chiquito | 1 | | 1 | | 4 | |
| | | El Salto | | | | | 3 | |
| | | Marraganti | 17 | | 8 | | 8 | |
| | | Quebrada Marraganti | | | | | 1 | |
| | | Boca de Riocito | 1 | | | | 1 | |
| | | Lajas Blancas | 19 | | | | 1 | |
| | | Las Peñitas | | | 1 | | 1 | |
| | | Nuevo Vigía | | | 5 | | 2 | |
| | | Playona | 1 | | 2 | | 2 | |
| | | Pueblo Tortuga | 2 | | 1 | | 2 | |
| | | Villa Caleta | | | | | 2 | |
| | | Quebrada Vigia | | | 2 | | 2 | |
| | Manuel Ortega | Boca de Tigre | | | 12 | | 2 | |
| | | Bajo Chiquito | | | 2 | | | |
| | | Canaan | | | 1 | | | |
| | | Corozal | 4 | | 4 | | 3 | |
| | | El Común | | | 1 | | 1 | |
| | | Lajas Blancas | | | 1 | | 1 | |
| | | La Esperanza | 10 | | 1 | | 1 | |
| | | La Pulida | 5 | | 2 | | | |
| | | Naranjal 2 | | | 4 | | 1 | |
| | | Nazareth | | | 1 | | 1 | |
| | | Pueblo Nuevo | | | 1 | | | |
| | | Punta Grande | 5 | | 1 | | | |
| | | Quebrada Muerto | | | 1 | | | |
| | | Río Chico | | | 2 | | | |
| | | Riocito | 1 | | 1 | | 2 | |
| | | Villa Nueva | | | 5 | | 3 | |
| | Cirilo Guaynora | Unión Choco | 2 | | | | 33 | |
| | | Puente | | | 1 | | | |
| Wargandi | Wargandi | Morti | 10 | | 20 | | 12 | |
| | | Nurra | 8 | | | | 1 | |
| | | Wala | 14 | | 13 | | 18 | |
| Total | | | 168 | | 146 | 26 | 144 | |
| Imported cases | | | | 2 | | 13 | | 2 |
| Grand Total | | | 170 | | 185 | | 146 | |

All the malaria cases were treated following the NMP treatment scheme and using antimalarial drugs specific to the infecting *Plasmodium* species. According to the malaria cases in general, the malaria-endemic belt was located within the "success zone" and present in most months with exception of December, January, February and March (where variations were observed in the zones of success and security) and the months of June and July (which crossed into the alarm zone). The endemic channel graphically presents the actual incidence of a disease over its historical incidence (Fig 5). The cases can be distributed according to its incidence: a) Exit zone: the number of notified cases presents a smaller than expected frequency.

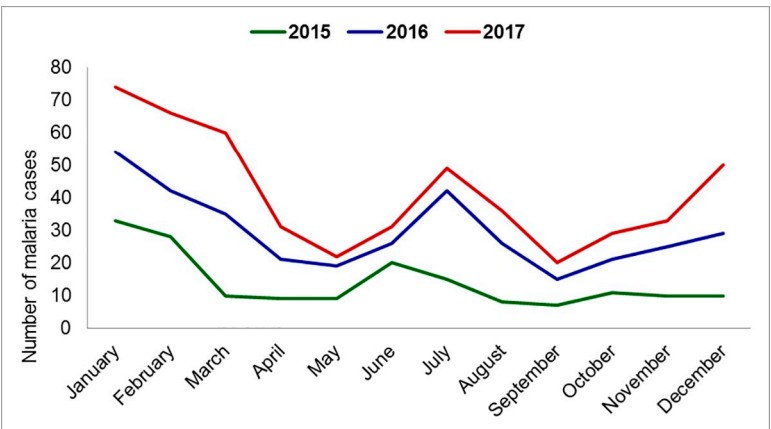

**Fig 3. Chronological distribution per month of malaria cases in the Darien Region, Panama. 20015 to 2017.**

b) Security zone: the number of cases presents a stable behavior; and c) Alarm zone: the number of notified cases presents a higher than expected frequency [38,39].

People between the ages of 1 to 59 years of age presented a higher malaria incidence (n = 455; 91.0%) in comparison to the rest of the age groups. The median age for malaria cases were 24.0; 23.0 and 25.5 years of age for 2015, 2016 and 2017 respectively. The average age of malaria-infected people were 26.26, 28.14 and 28.77 years old for 2015, 2016 and 2017 respectively, and these were not significantly different (one-way ANOVA with Tukey's multiple comparison test, 2015 vs. 2016: p value: 0.63; 2015 vs. 2017: p value: 0.48; 2016 vs. 2017: p value: 0.95). The age span registered for malaria cases ranged from one to 85 years of age. The distribution of malaria cases per age and sex group is detailed in Table 4.

### Diagnosis and treatment

In 2015, Darién registered a total of 170 malaria cases, of which 99.9% (n = 168) were caused by *P. vivax* and two cases by *P. falciparum* imported from Colombia. Gametocyte presence was observed in 42.4% (n = 72) of the cases based on microscopic parasite diagnosis. In 2016, there were 185 registered malaria cases, with 35 cases caused by *P. falciparum*, of which 13 were imported from Colombia and 33% (n = 61) of the cases were gametocyte positive. Finally, there were 146 malaria cases diagnosed in 2017 with 58.2% (n = 85) showing the presence of gametocytes and two imported *P. falciparum* cases from Colombia. In the period from 2015 to 2017, 43.5% (n = 2018) of the total positive samples presented gametocytes. The microscopy-based diagnosis presented mainly a moderate to high parasitic density (trophozoites, schizonts

**Table 3. Malaria surveillance indicators malariometrics in Darién Region, Panama, from 2012 to 2017.**

| Malariometric Indicators | 2012 | 2013 | 2014 | 2015 | 2016 | 2017 | National Level [2017] |
|---|---|---|---|---|---|---|---|
| | Population 52,368 | Population 53,025 | Population 53,690 | Population 54,366 | Population 55,055 | Population 55,753 | Population 4.098,135 |
| Cases | 275 | 138 | 205 | 170 | 185 | 146 | 689 |
| API | 5.3 | 2.6 | 3.8 | 3.1 | 3.4 | 2.6 | 0.2 |
| SPR | 1.1 | 0.6 | 1.0 | 1.0 | 1.4 | 1.3 | 1.8 |
| ABER | 48.1 | 45.8 | 39.9 | 30.5 | 24.7 | 19.6 | 1.0 |

**API**: total number of positive slides for parasite in a year x 1000/Total population, **SPR**: number of laboratory-confirmed malaria cases per 100 suspected cases examined, **ABER**: number of blood slides read over population at risk x 100 by year.

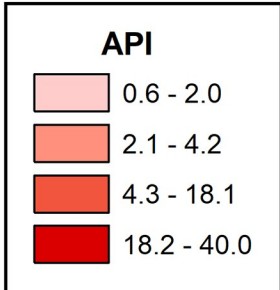
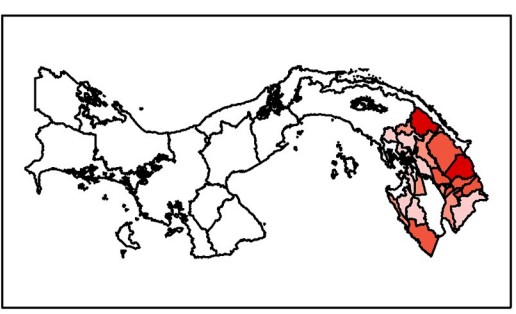
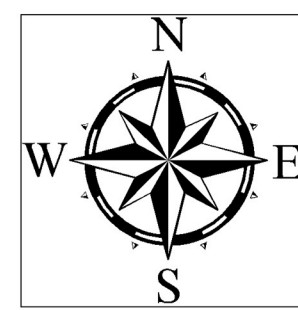

**Fig 4. Malaria distribution by corregimientos according to the API in Darién region 2015–2017. API**: total number of positive slides for parasite in a year x 1000/Total population.

and gametocytes). The highest parasitic density had more than 6,000 parasites/µl of blood (n = 13 cases; 2.6%), followed by 4,000–6,000 parasites/µl of blood (n = 39 cases; 7.8%) and 2,000–4,000 parasites/µl of blood (n = 93 cases; 18.6%). There were also parasitemias with moderate densities: 1,000–2,000 parasites/µl of blood (n = 121 cases; 24.2%) and 500–1,000 parasites/µl of blood (n = 101 cases; 14.5%) (Table 5). The patients were successfully treated according to national protocols for the therapeutic treatment of patients suffering *P. vivax* malaria.

The 100% of the patients presented malaria only once during the entire year. All treated cases were confirmed via microscopic diagnosis and patients were successfully treated according to the national guidelines for the treatment of *P. vivax* and *P. falciparum* patients. This included using 1,500 mg of CQ and 210 mg of PQ during seven consecutive days for *P. vivax* malaria. and Coartem® (20mg of Artemether + 120 Lumefrantine) during three consecutive days for *P. falciparum* malaria. Diagnostic procedures consisted in two methodologies: the RDTs and thick blood smear test. Microscopy of thick-blood smears continue to be the standard test for malaria diagnosis. According to the epidemiological research data conducted by the NMP, a total of five recurrent *P. vivax* cases were detected: one in 2015, another in 2016 and three in 2017.

## Signs and symptoms

According to the epidemiological findings, the main signs and clinical symptoms most frequently manifested in the malaria patients were fever, followed by chills, sweating and general malaise with *P. vivax* and *P. falciparum* respectively. Most of the cases presented fever as the most significant symptom (Table 6).

## Migration and malaria

The Darien region has always registered a migratory movement of people from the neighboring regions within Panama, such as Guna Yala and Madungandi. These migrants use roads,

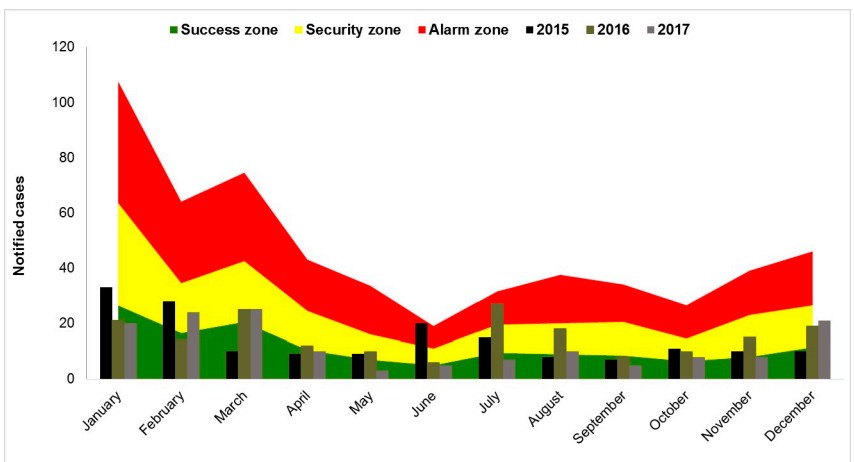

**Fig 5. Monthly endemic channel built from registered malaria cases between 2000 and 2014 in Darién region, Panama. Success zone**: the number of reported cases presents a lower frequency than expected. **Security zone**: the number of cases presents a stable pattern. **Alarm zone**: the number of reported cases presents a frequency that is higher than expected.

**Table 4. Distribution of malaria cases by age group and sex in Darién region, Panama.**

| Age Group | Number of cases | | | | |
|---|---|---|---|---|---|
| | Female | Male | Total cases by age group | % cases | % Cumulative |
| < 1 | 4 | 2 | 6 | 1.2 | 1.2 |
| 1–4 | 21 | 18 | 39 | 7.8 | 9.0 |
| 5–9 | 25 | 20 | 45 | 9.0 | 18.0 |
| 10–14 | 28 | 29 | 57 | 11.4 | 29.3 |
| 15–19 | 19 | 41 | 60 | 12.0 | 41.3 |
| 20–24 | 13 | 35 | 48 | 9.6 | 50.9 |
| 25–34 | 27 | 55 | 82 | 16.4 | 67.3 |
| 35–49 | 17 | 73 | 90 | 18.0 | 85.2 |
| 50–59 | 13 | 21 | 34 | 6.8 | 92.0 |
| 60–64 | 4 | 10 | 14 | 2.8 | 94.8 |
| 65 y + | 15 | 9 | 24 | 4.8 | 99.6 |
| NI | 1 | 1 | 2 | 0.4 | 100.0 |
| Total | 187 [37.3%] | 314 [62.7%] | 501 | | |

NI = No information about the age group

trails, the Panamericana road and rivers that communicate the different towns located within the Darien region. Furthermore, there is an active migratory movement within Darien due to economic, commercial and agricultural activities. Furthermore, there is migration of people coming from Colombia. According to the data obtained, a total of 19 malaria cases were imported from Colombia during the period of 2015 to 2017.

In addition to the imported cases from Colombian immigrants, we must consider the malaria cases from people that live in Darién, who might get infected in endemic areas along the border with Colombia. This is due to the constant migration that occurs in this zone because of economic, commercial and socio-cultural activities. These malaria cases are difficult to detect and quantify by the sanitary and migratory authorities once the person arrives to their community given that generally the patients deny having traveled to the Colombian side. The cities that occupy Darién, in the border with Colombia (Bajo Chiquito, Pucuro, Boca de Cupe, Paya, Alto Limon, Manene, Cocalito, El Guayabo, Jaque and Biroquera), are the main communities that maintain a constant migratory activity with the neighboring communities that exist in the endemic Colombian regions of Uraba and Costa del Pacifico (Fig 6). There is also the migratory movement of legal and illegal immigrants, displaced populations and drug traffickers who hamper the ability of sanitary and migratory authorities to control the migration and diminish their impact on malaria incidence of this region.

**Table 5. Characterization of malaria cases according to parasitic density and presence of gametocytes, following microscopic quantification as parasites/μl of blood.**

| Microscopy, Parasitemia range | No. of diagnosed cases | Cumulative frequency | % Diagnosed cases | No. cases whit gametocytes | Cumulative percentage |
|---|---|---|---|---|---|
| 1–100 | 61 | 61 | 12.2 | 12 | 12.2 |
| 100–500 | 73 | 134 | 14.5 | 28 | 26.7 |
| 500–1 000 | 101 | 235 | 20.1 | 49 | 46.8 |
| 1 000–2 000 | 121 | 256 | 24.2 | 53 | 71 |
| 2 000–4 000 | 93 | 449 | 18.6 | 49 | 89.6 |
| 4 000–6 000 | 39 | 488 | 7.8 | 17 | 97.4 |
| > 6 000 | 13 | 501 | 2.6 | 10 | 100 |

**Table 6. Most frequent malaria signs and symptoms observed in patients with *P. vivax* and *P. falciparum* in Darién region, Panama, from 2015 to 2017.**

| Signs and symptoms | *P. vivax* | Frequency (%) | *P. falciparum* | Frequency (%) |
|---|---|---|---|---|
| Headache | 327 | 67.6 | 12 | 70.6 |
| Fever | 459 | 95.0 | 16 | 94.1 |
| Sweating | 451 | 93.2 | 16 | 94.1 |
| Chills | 453 | 93.6 | 15 | 88.2 |
| General malaise | 448 | 92.6 | 16 | 94.1 |
| Vomiting | 139 | 28.7 | 5 | 29.4 |
| Diarrhea | 130 | 26.8 | 5 | 29.4 |
| Nausea | 141 | 29.1 | 5 | 29.4 |
| Appetite loss | 138 | 28.5 | 4 | 23.5 |

## Climate and malaria

An increase in the number of cases from the chronological distribution of malaria cases accumulated from 2015 to 2017 were observed from December to March and in the months of

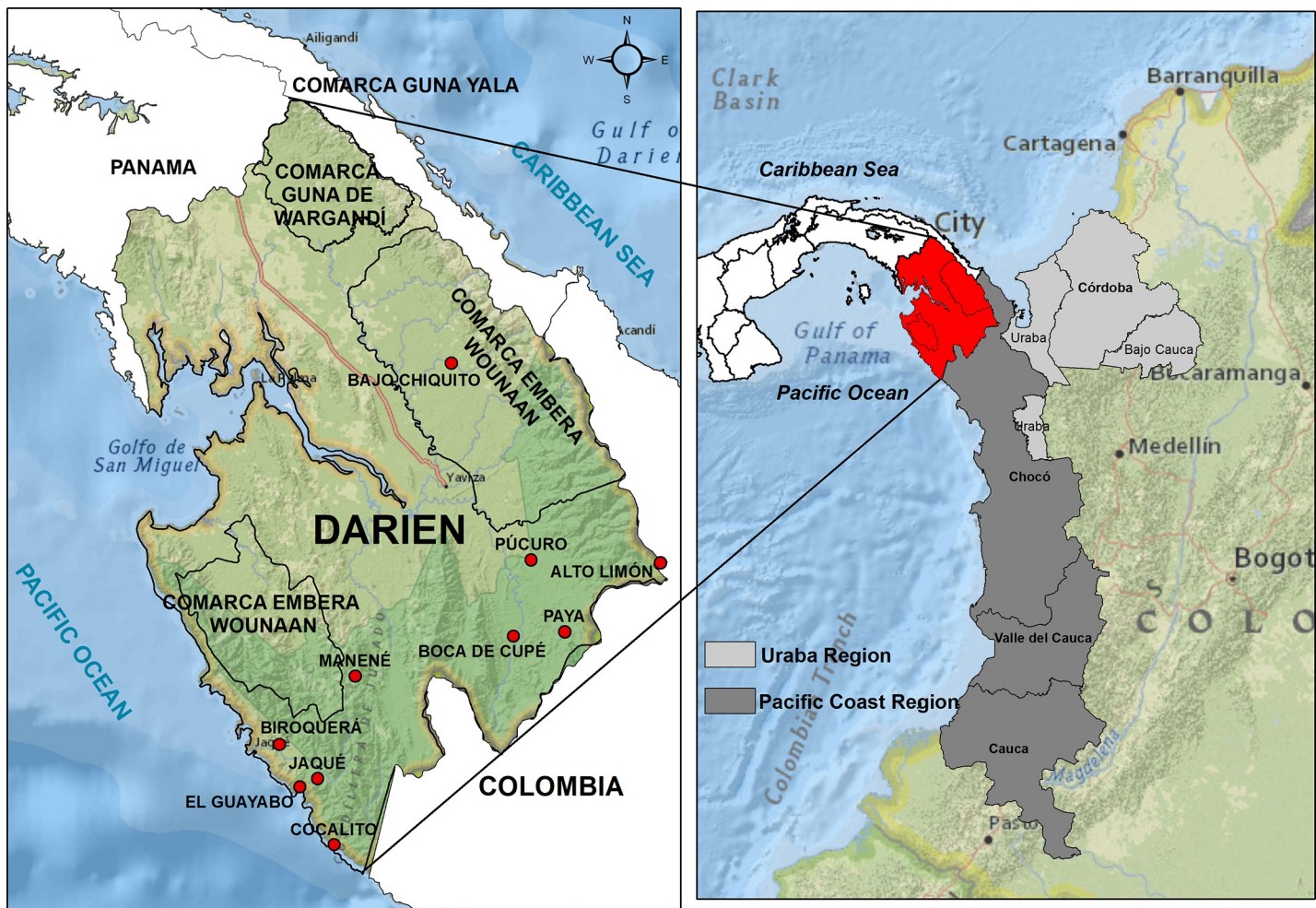

Service Layer Credits: Content may not reflect National Geographic's current map policy. Sources: National Geographic, Esri, DeLorme, HERE, UNEP-WCMC, USGS, NASA, ESA, METI, NRCAN, GEBCO, NOAA, increment P Corp.

**Fig 6. Geographical location of bordering communities in Darién region.** These communities maintain an active interaction and migration activity with malaria endemic regions of Colombia.

June and July. The month of January registered the greatest number of diagnosed malaria cases (74 cases; 14.8%), followed by the month of February (66 cases; 13.2%), march (60 cases; 13.2%), December (50 cases; 10%), and the month of July (49 cases; 9.8%). The months of January to April correspond to the dry season and from May to December to the rainy season (Fig 3).

## Malaria control measures used in the area

Based on the information gathered from the NMP of Darién, the antimalarial activities conducted were: indoor residual spraying of the organophosphate insecticide Sumithion® WP 40% against populations of Anopheline malaria vectors with an application frequency of every four months. The coverage of application in all malaria endemic localities was of approximately 90%. This reached 85% in non-indigenous communities and 25% in indigenous communities. Additionally, treatments via fogging with the insecticide deltamethrin with a portable fogging system ULV were conducted as soon as malaria cases or outbreaks were reported in the community. These applications had a frequency of application of three continuous days for three consecutive weeks. In addition, the microbial agent *Bacillus sphaericus* (Vectolex®) was applied to anopheline breeding pools. It is important to indicate that the NMP does not monitor regularly the susceptibility and efficacy of applied insecticides against mosquito vectors. Also, the NMP does not conduct research to learn the biting behavior of anopheline mosquitoes in the peri- and intradomicile areas.

The epidemiological surveillance was conducted in approximately 600 communities that exist throughout Darién. A total of 105 communities presented malaria cases during the period of 2015–2017 with an annual average of 35 communities. Surveillance in this communities were conducted every 15 days or once a month while those communities that did not present a malaria case were visited one to three times a year. Surveillance was based on the active and passive search, diagnostic procedures and treatment of malaria cases. The assessment was conducted taking into consideration the regional plan for malaria control, the intersectoral articulation, the intervention coverage, the access to diagnostics and malaria treatment, and the epidemiological surveillance.

## Discussion

### Malariometric indicators

The malaria indicators have the purpose of offering the epidemiological tools required to generate, analyze and utilize the information on malaria quantification, distribution and risk factors. Thus, it allows an adequate selection of intervention strategies and a more rational allocation of available resources for the surveillance and control of malaria [27].

The different regions of Panama present a considerable variation in malaria transmission incidence and risk of infection. We used the API [35] to estimate the incidence and the epidemiological impact of malaria in Darién. This indicator is used to epidemiologically stratify the malaria zones according to the risk level. Furthermore, it contributes to the evaluation and reorientation of the action plan, management and assessment of public health policies directed towards combating malaria. Darien presented APIs of 3.1/1000, 3.4/1000 and 2.6/1000 inhabitants between 2015 and 2017 respectively. This represented an average API of 3.0/1000 inhabitants characterizing this region as one with a moderate risk of transmission. Moreover, this incidence level represents an increase in transmission with respect to that registered in 2014 (API = 0.9/1000 inhabitants).

Considering the number of cases per month during the 2015–2017 period and according to the endemic channel, most months registered cases within the success and security zones. This

indicates that malaria was maintained with a lower number of cases than expected. Hence, we can consider that malaria transmission during this period was stable and outside of the epidemic level [38]. The endemic channel is a tool of ample use in epidemiological surveillance systems that allow the description of a disease and the assessment of its endemic or epidemic nature [39]. The average SPR was 1.2%, which was similar to the national average (1.8%). The SPR indicates the degree of positivity and the surveillance efficiency using thick-blood smears for diagnosis and does not have a direct correlation to the API. The lower the SPR, the greater the epidemiological surveillance efficiency. The ABER registered a high average of 24.9%, indicating that a great number of blood samples were collected in search of suspected cases. Through the ABER we can measure the intensity of the epidemiological surveillance and it is necessary that the blood samples taken in the active and passive search be more selective and specific for people living in localities with a level of transmission risk.

Malaria was found across all age groups, nevertheless, it is interesting to note that a high percentage of cases were seen in the age group 1–14 years of age (n = 141, 28.1%). Despite the high number of cases in this age group we did not observe complicated malaria cases. This high number also suggests that the transmission was both intra and peridomicile. Similar studies have indicated that the malaria transmission risk was greater for children younger than 10 years of age and diminished with increasing age [40,41]. It is important to remember that this population is considered vulnerable and with greater chances of health complications [42]. We observed a small predominance of malaria cases in males (62.7%) in comparison to females. This predominance could be related to the behavior of male inhabitants, who usually migrate to their work sites at the early hours of the night and whose habits of nocturnal recreation favor malaria transmission. Similar studies have indicated that it is expected that males will have a higher rate of malaria cases than females due to their tendency to visit forested areas or areas with active mosquito breeding, in addition to the aspects mentioned above [43,44]. In addition, our results, agree with other similar findings [45,46] in that the more economically active age groups (15–64 years of age) registered a high malaria incidence (n = 328, 65.5%). In addition, a study conducted in Brazil, indicated that in areas with active malaria transmission along the border, like in the Darien region, the migration of infected people affected malaria-free regions [47,48]. These malaria-free regions are assumed to have a population where all age groups are equally vulnerable to malaria infection [49,50].

## Migration and malaria

Malaria in the Darién region is affected by diverse factors that vary in time and space. One of the main variables that determine malaria incidence, within the transmission risk factors, is the migration of symptomatic and asymptomatic people from one community to another within a region or from other endemic regions. Among the main endemic regions in Panama are the indigenous comarcas Guna Yala and Madungandi that migrate using the Panamericana highway and through the rivers Chukunaque, Tuira, among others, to travel to the main cities of Santa Fe, Meteti, Yaviza, La Palma, and the comarcas Emberá—Wounaan and Wargandi. The migration also occurs to other small communities located along the different rivers and roads that communicate these communities, dispersing malaria within the Darién region and to other regions in Panama.

Historically, the population movement in Darién has represented a weakness and challenge to the NMP efforts to successfully control malaria. The internal migratory movement in Darién is characterized by the presence of different indigenous populations, for whom migration is part of their culture, as well as the general population movement tied to socio-economical activities. Another important migratory factor is the fact that Darién is in the border region

with Colombia and experiences a challenging migration control with legal and illegal immigrants moving from South America in their way to the northern hemisphere. The Panamanian authorities lack installations, enough personnel and logistic resources to detect migrants with malaria symptoms. This is worsened by the activity of drug traffickers and the movement of illegal immigrants and displaced populations. This situation deteriorates even more due to the ecological, and environmental conditions. This is a mountainous forested area with different border points from where immigrants can cross, hampering efforts by migrant authorities and the NMP to control migration and diminish its effect on malaria incidence in this region. Hence, despite the permanent malaria surveillance stations in Jaque (Pacific coast) and Boca de Cupe (border of the Tuira river), and the monthly malaria surveillance in the border communities of Paya, Pucuro, El Guayabo and Cocalito, it is not sufficient due to the lack of resources. In addition to the communities mentioned above which are used as monitoring surveillance points, there are other communities such as Bajo Chiquito, Biroquera and Manene, which also maintain cultural, commercial, economic and family relationships with Colombian communities located in two endemic regions. These regions are Uraba-Bajo and Cauca-Alto region and the Costa del Pacifico region which have contributed annually, with 60% and 10–30% respectively, of malaria cases in Colombia in the last 50 years [51,52].

Disease transmission in border regions is notoriously difficult to manage, due to the legal and illegal movement of people and goods, cultural differences, border conflicts, and differences in national public health regulations [53]. The movement of people has traditionally contributed to malaria propagation [54]. Epidemic outbreaks usually occur when there is a combination of three factors: migration of people from endemic areas or regions, presence of a competent vector and a susceptible population group [55]. Evidence from prior worldwide malaria eradication programs [56], as well as from recent control and elimination efforts [57,58] highlight the need to consider human movement in the design of elimination strategies. Absence of this factor contributed to the failure of the earlier malaria eradication campaigns in the 1950s and 1960s [59]. Recently, following the revival of the malaria elimination paradigm [60], it has been recognized that despite the importance of migratory dynamics, there are scarce attempts in the national politics of most countries at introducing strategies dealing with this migratory movement with respect to malaria control [61].

The transit of infected people from South America or Africa, where drug-resistant *P. falciparum* have been confirmed [62,63], through Darién, can increase the risk of new malaria outbreaks. This is extremely likely due to the vulnerability and receptivity of the communities and the existence of anopheline vectors in this region. In Colombia, a total of 83,356 malaria cases were registered in 2016, from which 47,497 (57.0%) were caused by *P. falciparum*, 33,055 (40.0%) by *P. vivax*, and 3.3% mixed infections [64]. The magnitude of the contribution of imported malaria cases to the total number of cases in different endemic regions, like Darién, will depend on socioeconomic, cultural and health-related factors as well as environmental conditions that determine the migratory movement [65]. It is important to indicate that the proportion of imported cases in relation to the autochthonous cases can vary with time. These variations can be associated to changes in the frequency and patterns of the migrant population. Studies conducted in Colombia indicate that the variations have a seasonal pattern related to labor migration [66]. Nevertheless, our study did not determine the patterns of movement of the migrant population that transits in Darién. In addition to the socioeconomic and cultural factors, this could explain the variations in the proportion of imported cases and the appearance (in time/space) of malaria outbreaks in Darién. This is especially true if we consider the large movement of people and the periodical variations in transmission levels within and outside Darién.

## Climate and malaria

Darién has in general a climate with rainy precipitations from May to December, with the highest precipitations in the months of May, October and November [26]. This creates different types of mosquito breeding habitats and leads to an increase in the anopheline vector population. The diversity of habitats, in turn, creates a challenge for mosquito control strategies and is crucial for the maintenance of malaria transmission. The chronological distribution of malaria shows constant disease transmission with an increase in certain months of the year. The highest number of cases occurred in November, which is associated with the rainy season, agricultural and commercial activities and the migratory movement. In general, the highest intensity of malaria transmission occurs during and immediately following the rainy season. As it is the case with the rest of the country, transmission during the period between the rainy and dry season (December to January) is maintained due to the development of temporary mosquito breeding pools that maintain the anopheline population.

A study conducted in the comarca Guna Yala, indicated that malaria transmission occurs throughout the year, with the highest number of cases occurring during the last three months. This is also associated with the rainy season, agricultural and commercial activities, and the migration that increases at the end of the year [17]. Meanwhile, studies conducted in the *comarca* of Madungandi indicated a clear association between El Niño Southern Oscillation (ENSO) and malaria incidence in this region [67]. A similar study showed that ENSO, rainfall and NDVI (normalized difference vegetation index) were associated with the number of malaria cases in Guna Yala during the study period (1998–2016). This indicates a need for further studies that evaluate weather impacts on malaria vector ecology, as well as the association of malaria vectors with Gunas, paying attention to their socio-economic conditions of poverty and cultural differences as an ethnic minority [68].

Hence, we can assume that a higher malaria incidence occurs because of temperature, humidity and moderate precipitation (which increases the vector life span), in conjunction with biological and social human factors [69]. Understanding the role that temperature and precipitation play in malaria transmission is critical given that such factors can influence the development, geographic distribution, longevity, and oogenesis rate of the mosquito vector. Thus, such factors can have a tremendous impact in the dynamics of malaria transmission [70–73]. Because of these reasons, malaria has been considered a disease that is prone to be affected by global climate change [70]. Therefore, an integrated approach, one that considers climate as a factor along the other aforementioned factors, is needed in the fight against malaria. This would allow the exploration of current and future risks and vulnerabilities in the transmission of endemic/epidemic malaria [74].

## Signs and symptoms of malaria infection

Taking into consideration that human malaria can show a vast clinical spectrum that includes asymptomatic infection, uncomplicated, complicated and lethal malaria; and given that it depends on a complex interaction between the parasite, the human host and the environment, we decided to assess the main malaria symptoms in patients.

Researchers and clinicians have established diagnostic criteria based on the clinical manifestations upon disease onset, which has aided in forming an integrated approach to improve the management and treatment of severe malaria [75]. From the epidemiological research of malaria cases, we were able to determine that the most frequent symptoms associated with *P. vivax* and *P. falciparum* parasitemia were fever and chills. No significant differences were observed between infections caused by *P. vivax* or *P. falciparum* in terms of these two main symptoms. No complicated malaria cases were observed during this study. It has been shown

that in endemic areas or regions with high malaria transmission, such as Darién, the population develops a level of immunity against the parasite, which in turn changes the clinical manifestations of the disease. Infection by any of the *Plasmodium* species can result in fever, along a series of important signs and symptoms, which are not exclusive of malaria infection. Malaria clinical manifestations are more frequent in children and young people, while the asymptomatic infections are more common in adults [76,77]. The main malaria infection symptoms are fever, chills, sweating, and sometimes accompanied by headaches, vomiting, diarrhea, general malaise and myalgias. In endemic zones, fever is the most detectable symptom for a possible malaria diagnosis, after excluding other causes [78]. Fever can have a cyclical periodicity or present an irregular pattern [79,80]. Infections by *P. vivax* are characterized by normally having a low lethality and benign course, although a few complicated malaria cases have been described for this parasite [81].

Through its efforts, the NMP in Darién has determined that there are cases of people with unapparent malaria, who tend to develop a degree of immunity. These people become a significant source of gametocytes for the local anopheline mosquitoes. A different behavior has been observed in malaria infected people coming from non- endemic malaria areas, who tend to present intense malaria symptoms with low parasitemia and in which all age groups have the same risk of developing clinical manifestations and complicated malaria forms [78]. It has been shown that the immunological state of the human host is another determining factor in the clinical manifestations of malaria. This is especially true for adults residing in geographical zones classified as having moderate or intense transmission. With time, the individuals that are exposed to malaria in this region acquire a certain level of protection against the infection. This reduces the risk of developing a complicated form of the disease but without conferring a complete protection against malaria. Early malaria symptoms vary and are not malaria-specific, common but equally life-threatening infections such as febrile ailments and viral and bacterial infections possess similar signs and symptoms to malaria; hence, clinical diagnosis is quite challenging and unreliable [82]. No differences were observed in the frequencies of signs and symptoms between patients with *P. vivax* and *P. falciparum*.

## Diagnostics

In this study we were able to determine that 100% of all thick blood smears that were malaria positive were diagnosed as such via microscopy. We were able to establish from the NMP observation of blood samples, diagnostics and epidemiological research that the time interval between the start of symptoms, blood test and the parasitological diagnosis was from 1 to 20 days. This further depended on the sample origin, which in turn could be one of the factors that contribute to malaria transmission in regions where RDTs are not used. It is important to reduce the diagnostic time so that we can avoid having infected people becoming a source of infection for new malaria hotspots. The timely diagnosis and early treatment of malaria cases are of vital importance to diminish the disease duration, period of malaise, complicated cases, mortality, and patient inactivity. They will also contribute by interrupting the cycle of transmission [83]. The microscopic continue to be the "gold standard" for parasite diagnosis [84]. In 2009, WHO established that the parasitological diagnosis via microscopy and the rapid diagnostic test should be necessary for the management of malaria. It also stipulated that empirical treatments should only be done in regions where these practices are not available [85,86].

*Plasmodium vivax* infections accounted for a high percentage (92.2%) of diagnosed cases in Darién between 2015 and 2017. In studies previously conducted in Panama, *P. vivax* was associated with the highest frequency of malaria in all the evaluated localities. *Plasmodium*

*falciparum* showed lower incidence and contribution to cases of malaria in the target population [17,19]. A high gametocyte incidence average (43.5%) was observed at this same time. The moderate and elevated gametocyte densities and high gametocyte incidence could be associated to the increase in the transmission risk and disease maintenance in the region. It is important to consider that infections with a high gametocyte density are generally more infectious [87]. Based on this, our study indicates that the surveillance system used to identify suspected cases are being conducted late in Darién. Studies have shown that malaria infections that persist for long periods of time provide more opportunities for an infected person to serve as a source of infection for mosquitoes feeding on a gametocyte-laden blood. Furthermore, the existence of asymptomatic people with persistent infection and prevalence of gametocytes has been reported in regions with high and moderate transmission [88]. Gametocytes from *P. vivax* are more prevalent before a recurrence or progression of a primary infection than those deriving from *P. falciparum* [89,90]. In this manner, most patients have sufficient gametocytemia to allow transmission before the disease is diagnosed or treated [91,92]. In addition, *P. vivax* gametocytes are transmitted more efficiently by *Anopheles* mosquitoes than those deriving from *P. falciparum*. They are also transmitted at lower parasitic densities [93,94].

The majority of malaria cases registered moderate (500–1,000 parasites/μl of blood) to high parasitemia (4,000–6,000 parasites/μl blood). This could be due to the late detection of malaria suspected cases. The determination of parasitemia levels is important because they are related to malaria severity and serve to evaluate the patient's therapeutic response to different levels of parasitemia intensity. Individuals living in endemic areas who have been exposed to the parasite from an early age display a certain degree of immunity, as exemplified by low parasitemia levels when exposed to new infections [95,96]. The parasitemia density in patients with acute *P. vivax* infections depends on many factors, including the age, semi-immunity condition, delays in treatment, self-medication and a range of factors associated with the host and the parasite [97]. The parasite count in *P. falciparum* infected patients is of critical importance for the clinical decisions and management of the disease [98]. It is important to indicate that in endemic regions such as Darién, in contrast to low endemic areas, malaria-infected people can acquire a certain level of immunity and also present low levels of parasitemia in comparison to non-immune subjects, such as travelers or immigrants. This could lead to a decrease in the diagnostic sensitivity in this group of patients. Given that Darién is characterized as being endemic, we could expect to see cases with low parasitemia in people who have had prior malaria infections, due to the potential acquisition of immunity. On the other hand, parasitemia can be high in the young population that acquired malaria for the first time. However, further studies regarding the association between previous episodes of malaria and parasitemia levels are needed. In addition to the thick blood smear and microscopic diagnosis, one of the tools and strategies that have considerably helped the timely malaria diagnosis in Darién is the use of RDTs. Since their implementation at the end of 2016, RDTs have allowed the early treatment and elimination of malaria parasites in patients. In this way, it has decreased or interrupted disease transmission and the appearance of new malaria outbreaks.

## Antimalarial treatment

The patients diagnosed with malaria were successfully treated according to the national guidelines on the therapeutic treatment of *P. vivax* and *P. falciparum* malaria patients. All adult and young patients diagnosed with *P. vivax* malaria received a radical antimalarial treatment according to the NMP treatment scheme [27]. The coverage and fulfillment of the treatment scheme with CQ and PQ was 100%. The objective of the chemical-prophylactic treatment is the elimination of clinical signs and symptoms, reduce the immediate risk to the host,

eradicate peripheric asexual parasitemia, get parasite clearance, prevent relapses and interrupt the transmission cycle [99]. Five malaria recurrent cases were detected during this study, one case each in 2015 and 2016 and three in 2017. Once the recurrent cases were detected, they were successfully treated using the *P. vivax* treatment scheme and each case was closely monitored. The capacity of *P. vivax* to form hepatic latent stages (hypnozoites), which can cause recurrent infections weeks or months following the initial infection, constitutes an important challenge for the complete eradication of this parasite in patients. No current medication attains all these objectives; hence, it is required to use a combination of antimalarial drugs that target specific key life stages in the parasite [100].

According to the NMP, in the last five years there has been an annual average of 5% (n = 36) in *P. vivax* recurrences from all the malaria-diagnosed cases [101]. Given the low frequency of recurrent cases in Darién, we can assume that they do not have a significant impact in malaria maintenance, considering that the cases were identified and treated in a timely manner according to the malaria treatment schemes [27]. Previous studies have compared the efficacy of the different treatment schemes using PQ to prevent *P. vivax* recurrence [102]. Although, the standard PQ regime (0.25 mg/kg/day for 14 days) has a significant incidence of relapses, it is the most commonly used treatment around the world [103]. It is important to indicate that currently in Panama there are no studies or evidence that shows a therapeutic failure when using CQ and PQ to treat *P. vivax* infections. In addition, there are no studies that determine if the new malaria episodes derive from a recurrence (via hypnozoite activation), from a reinfection originating from a new infectious mosquito bite or from the parasite resistance to PQ or CQ. Hence, it is imperative that studies are conducted in this area to determine the origin of the recurrent cases.

Studies conducted in endemic regions, recurrent *P. vivax* infections can originate due to three situations: 1) parasite recrudescence in the blood, which can be caused by resistance to antimalarial treatments, administration of an inadequate dose, or due to suboptimal drug absorption; 2) reinfection due to new infectious mosquito bites; 3) relapse due to reactivation of liver hypnozoites [104]. The diagnostic methods by themselves cannot distinguish between these three types of recurrence. Nevertheless, a complete study that includes supervised treatment, therapeutic efficacy through individual follow-ups, microscopic and molecular diagnostic, blood drug level quantification and the parasite genetic characterization can reveal the origin of the recurrent cases [103,105,106].

To eliminate *P. vivax* malaria in Panama, it is necessary to maintain a surveillance of the malaria recurrent cases in all endemic regions. This kind of surveillance should also include studies determining the genotype of the circulating *P. vivax* and its associated factors. This type of focus not only allows the classification of the recurrent malaria cases but also allow us to determine whether a given case is a reinfection, due to therapy failure, or a resistance to the antimalarial drugs. Furthermore, this opens the possibility of identifying Primaquine (PQ) tolerant markers, and the evaluation on the efficacy and safety of therapeutic PQ treatments with optimal doses for different epidemiological contexts. This would also allow the search and evaluation of novel therapeutic agents against the hepatic hypnozoites [107].

## Assessment of control measures

Since the NMP establishment in 1956, the main anti-malaria strategies to reduce malaria transmission have been the search of suspected cases, its diagnosis and timely treatment for a radical cure, and the frequent IRS applications to control the population of anopheline vectors. It is well known that the interventions such as the ones previously indicated, tend to be highly successful in decreasing the prevalence and mortality in many countries. Nevertheless, its

effectiveness is threatened by the development of insecticide resistance and anti-malarial drug-resistance in mosquitoes and *Plasmodium* parasites, respectively.[108; 109].

During the study period, all the 105 localities that registered malaria cases were treated with indoor residual spraying (IRS) of fenitrothion and deltamethrin, applied via thermo-sprayers at the moment that an outbreak was registered. In treatments like this, it is necessary to conduct wall bioassays to determine the efficacy and residual activity of insecticides in different surfaces, as well as susceptibility test to determine if there is resistance to the applied insecticides. It is important to indicate that there is no current information on the susceptibility of the main malaria vector *An. (Nys) albimanus* to the applied insecticides. Insecticide resistance is a problem of great concern and it needs to be monitored in order to maintain the efficacy of vector control operations in the field. This is even more imperative given that countries in the Americas have decided to reduce the malaria morbidity by 40%, by the year 2020, taking as a base the official figures of 2015 [110]. An evaluation of the relationship between insecticide resistance intensity and the control strategy failure has only recently been recommended [111,112]. IRS failure has been observed when the levels of resistance intensity of the local vector populations are high [113].

NMP actions should include frequent evaluation of intra and peridomicile biting behavior of *An. (Nys) albimanus* to ensure the effectiveness of IRS use. The percentage of intra and peri-domicile human exposure that occurs are, consequently, important indicators of vector behavior that malaria control programs should examine for a more integrated management of malaria transmission. [114]. The persistence and predominance of intradomicile feeding behavior by populations of anopheline vectors that then become exposed to a high IRS coverage [115] suggests that there is still considerable space to improve technologies that target mosquitoes that enter households [116]. It is also necessary that the NMP monitors the development of insecticide resistance in *An. albimanus* populations.

In the last few years, malaria endemic countries have implemented an efficacious vector control program using two tools: IRS and long-lasting insecticide-treated bed nets (LLINs). These were used in combination or individually along with other strategies such as RDTs and combination therapies using artemisinin (ACTs). These approaches have resulted efficacious in the fight against malaria, with significant reductions in the morbidity and mortality associated to malaria [117,118]. To date, the NMP does not use LLINs. It is recommended that LLINs be used as an alternative to decrease malaria transmission.

Furthermore, it is also imperative that the NMP evaluates periodically the efficacy of anti-malarial drugs against *P. vivax* and *P. falciparum*. In the Americas, CQ is still used widely for vivax malaria. Although there are reports of CQ resistance in the Brazilian Amazon [119] where the majority of malaria cases and especially hospitalizations are due to this type of malaria, CQ is believed to remain quite effective. Although not conclusive, one of these studies suggests a possible association between anemia and CQ resistance [120]. Regular drug resistance surveillance is suggested, although reliable assays based on genotyping are not yet available due to the lack of a validated molecular marker associated with CQ resistance in *P. vivax*. Resistance of *P. falciparum* to antimalarial drugs is one of the most serious challenges facing national malaria control programs in the Americas. At present, *P. falciparum* is resistant to both chloroquine (CQ) and sulfadoxine-pyrimethamine (SP) throughout the Amazon Basin and to CQ alone on the Pacific Coast of South America [121].

Finally, to conduct frequent and effective interventions that control and eventually eliminate malaria in this region, it is necessary to strengthen the NMP intervention measures in the areas of logistics, technical-operative personnel, structural and financial support. These actions will maintain an adequate response to the current transmission conditions. Failing to act on these areas will lead to a weakening of the operational activities, and loss in quality, coverage, intensity and frequency that are necessary to effectively control malaria.

## Conclusion

This study provides an additional perspective on malaria epidemiology in Darién. Additional efforts are required to intensify malaria surveillance and to achieve an effective control, eventually moving closer to the objective of malaria elimination. At the same time, there is a need for more eco-epidemiological, entomological and migratory studies to determine how these factors contribute to the patterns of maintenance and dissemination of malaria. To achieve an effective control of malaria, and its possible elimination, it is necessary to strengthen the intervention measures conducted in this region by the NMP. This should be done at the logistical, technical, structural and financial level to maintain an adequate response to the current conditions that favor transmission. To prevent the recurrence of malaria outbreaks in this region, it is necessary to conduct a multidisciplinary and integrated malarial control management in conjunction with local and cross-border health authorities of Panama and Colombia. Furthermore, given that this area is home to three indigenous ethnic groups, it is necessary to establish health policies that are respectful of the cultural tradition and values of these indigenous communities. This intercultural approach could guarantee the sustainability of antimalarial measures. Lastly, it is necessary to conduct new studies that include epidemiological and entomological factors, and that characterize the migratory patterns along a longer period of time to determine the malaria frequency, distribution and risk factors for this region. This would provide much needed information allowing us to develop more effective surveillance, prevention and malaria control strategies.

## Acknowledgments

We would like to give special thanks to Dr. Panama Garcia, Director of Darién Health Region, and Santos Vega, Vector Control coordinator from Darién Health Region. We would also like to acknowledge the Director of the Department of Vector Control, Fernando Vizcaino, vector control technicians: Mario Avila, Silvio Betancurh, Eliberio Lopez and Regino Cordoba, research assistant Randhy Rodriguez and Alberto Cumbrera for the development of maps used in this study.

## Author Contributions

**Conceptualization:** Lorenzo Cáceres Carrera, Carlos Victoria.

**Data curation:** Lorenzo Cáceres Carrera, Carlos Victoria, Jose L. Ramirez, Carmela Jackman, José E. Calzada, Rolando Torres.

**Formal analysis:** Lorenzo Cáceres Carrera, Jose L. Ramirez, Carmela Jackman, José E. Calzada, Rolando Torres.

**Funding acquisition:** Lorenzo Cáceres Carrera.

**Investigation:** Lorenzo Cáceres Carrera, Jose L. Ramirez, Carmela Jackman, José E. Calzada, Rolando Torres.

**Methodology:** Lorenzo Cáceres Carrera, Carlos Victoria.

**Project administration:** Lorenzo Cáceres Carrera, Carlos Victoria, Rolando Torres.

**Software:** Carlos Victoria.

**Writing – original draft:** Lorenzo Cáceres Carrera, Carlos Victoria.

**Writing – review & editing:** Lorenzo Cáceres Carrera, Carlos Victoria, Jose L. Ramirez, José E. Calzada, Rolando Torres.

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
