## [Decision Letter · Decision Letter 0]

5 Jul 2019

PONE-D-19-14773

Study of the epidemiological behavior of malaria in the Darien Region, Panama. 2015 - 2017. Epidemiology of malaria in Darien

PLOS ONE

Dear  Dr Cáceres,

Thank you for submitting your manuscript to PLoS ONE. After careful consideration, we felt that your manuscript  requires substantial revision, following which it can possibly be reconsidered. As presented, the MS includes an excessive number of pages, references and figures. As quoted by the reviewers, It should be revised properly. Study design should be clarified and objectives need to be more realistic (according to the results). We therefore invite you to submit a revised version of the manuscript paying close attention to the specific points raised by both reviewers. 

We would appreciate receiving your revised manuscript by  August 10. To enhance the reproducibility of your results, we recommend that if applicable you deposit your laboratory protocols in protocols.io, where a protocol can be assigned its own identifier (DOI) such that it can be cited independently in the future. For instructions see: http://journals.plos.org/plosone/s/submission-guidelines#loc-laboratory-protocols

We look forward to receiving your revised manuscript.

Kind regards,

Luzia Helena Carvalho, Ph.D.

Academic Editor

PLOS ONE

Journal Requirements:

2. Thank you for including your ethics statement:  "This research determines the

epidemiology of malaria in Darién and is a concerted effort with the consent, approval

and participation of authorities, professional and technical personnel from MINSA and

ICGES.".  

- Please amend your current ethics statement to include the full name of the ethics committee/institutional review board(s) that approved your specific study.Once you have amended this/these statement(s) in the Methods section of the manuscript, please add the same text to the “Ethics Statement” field of the submission form (via “Edit Submission”).

- Please amend your current ethics statement to confirm that your named institutional review board or ethics committee specifically approved this study.

3. Please provide additional details regarding participant consent. In the ethics statement in the Methods and online submission information, please ensure that you have specified (1) whether consent was informed and (2) what type you obtained (for instance, written or verbal). If your study included minors, state whether you obtained consent from parents or guardians. If the need for consent was waived by the ethics committee, please include this information.

4. We noticed you have some minor occurrence(s) of overlapping text with the following previous publication(s), which needs to be addressed:

https://doi.org/10.1186/s12936-018-2235-3

https://doi.org/10.1155/2018/3954717

https://doi.org/10.1038/s41598-018-23801-9

https://doi.org/10.1155/2018/9163543

In your revision ensure you cite all your sources (including your own works), and quote or rephrase any duplicated text outside the methods section. Further consideration is dependent on these concerns being addressed."""

5. We note that Figures 1, 2, 4 and 6 in your submission contain map images which may be copyrighted. All PLOS content is published under the Creative Commons Attribution License (CC BY 4.0), which means that the manuscript, images, and Supporting Information files will be freely available online, and any third party is permitted to access, download, copy, distribute, and use these materials in any way, even commercially, with proper attribution. For these reasons, we cannot publish previously copyrighted maps or satellite images created using proprietary data, such as Google software (Google Maps, Street View, and Earth). For more information, see our copyright guidelines: http://journals.plos.org/plosone/s/licenses-and-copyright.

a)    You may seek permission from the original copyright holder of Figure(s) [#] to publish the content specifically under the CC BY 4.0 license.

6. We note that Figure 7 includes an image of a participant in the study.

Reviewers' comments:

Reviewer's Responses to Questions

**Comments to the Author**

1. Is the manuscript technically sound, and do the data support the conclusions?

Reviewer #1: No

Reviewer #2: Partly

2. Has the statistical analysis been performed appropriately and rigorously? 

Reviewer #1: No

Reviewer #2: No

3. Have the authors made all data underlying the findings in their manuscript fully available?

Reviewer #1: Yes

Reviewer #2: No

4. Is the manuscript presented in an intelligible fashion and written in standard English?

Reviewer #1: No

Reviewer #2: Yes

5. Review Comments to the Author

Reviewer #1: I thank the authors for a very comprehensive looking report they have produced on the epidemiology of malaria in the Darien region of Panama for the period: 2015- 2017. The article reports the case burden from P. falciparum and P. vivax malaria, the effectiveness of the first line therapies against these diseases measured by the prevention of subsequent recurrences. However, there are several areas which the authors can improve upon or better communicate the study findings.

Major comments:

1. Objectives and presentation of results

The authors state two objectives: 1) to determine the eco-epidemiological factors associated with the frequency and distribution of malaria transmission in Darién region; and 2) to evaluate the anti-malaria efforts by the NMP in influencing the control or reduction of malaria incidence in Darién.

• Regarding the latter objective, it is unclear from the manuscript what control measures were used by the NMP and how have the authors have evaluated the efforts of the NMP? This requires clarification.

2. Materials and Methods:

Study design: “We designed an observational, descriptive, retrospective and cross-sectional study to evaluate the main factors associated with the incidence, prevalence and distribution of malaria in the Darién region during the period of 2015 – 2017”

I have difficulty in understanding the study design, particularly regarding the labelling of this study as “retrospective and cross-sectional”.

• While the collation and analysis of data is retrospective, the design is not strictly retrospective like in a case-control design (the participants are asked questions regarding their exposure status retrospectively).

• The authors claim this is a cross-sectional study. However, the analysis uses the population size of the Darien region as its denominator when estimating the API and other malaria indicators. This uses the whole population of the Darien region (not only a cross-section of the study site). Perhaps, the authors were thinking that this data covers only a section of time (a 2 years period) rather than the cross-section of the population? If the population size was not used as denominator for the calculation of the indicators, then please clarify.

The study design does need a better description for the readers to fully understand the data. From what I understood from the description of the study details, I think this is a retrospective analysis of a prospectively collected data from routine monitoring/ public health surveillance.

The authors clearly state that “Malaria is a required notifiable disease in Panama”. This suggests that this is possibly “active surveillance study”.

3. Tables: 4, 5, 6 and throughout the manuscript

Please describe the method (or give reference to a specific command used in any software used) used for generation of the 95% confidence interval. Please indicate this in the material and methods section.

a. I do not understand what has been presented as confidence interval in these tables. I have calculated the estimate and corresponding 95% CI. While the point estimates agree, I do not agree with the confidence interval that has been presented.

Here is my code used to generate the percentage and 95% CI using open-source R software for calculating the 95% CI for Table 4. Even taking different statistical methods which can be used (the code below is using “Wilson’s method), I disagree with the 95% Confidence intervals presented, even taking the methodology used for this estimation into account.

# R version 3.6.0 (2019-04-26)

require(binom)

binom.confint( c(6,39,45,57,60,48,82,90,34,14,24,2),501, method="wilson")

If the authors have used any other method for estimating these 95% CI, then it needs a clear description in the methodlogy.

 

Minor comments

2. Abstract: “Males constituted 62.7% of the total cases, with 62.7% belonging to the economically active population”

It is not clear to which group is the latter 62.7% is referring. Is it 62.7% of the males who had malaria were economically active or 62.7% of the total cases were economically active?

3. Introduction: paragraphs 1-4

Define: “Autochthonous” when it is first appears in line 7 of Introduction section.

Define: Measoamerica/anthropogenic when they first appear in the text

Spell out NMP when it first appear

Delete superfluous full stop on line 3 of the fourth paragraph

Make it clear that comacrcas (and other ethnicity) is ethnicity

4. Define what annual parasitic index (API), slide positivity rate (SPR) and annual blood examination rate (ABER) mean. It is not clear to me how these were calculated.

5. Results: Malaria indicators

The authors report: “People between the ages of 1 to 59 years of age presented a higher malaria incidence (n = 455; 91.0%)”.

Higher compared to whom/to which age group?

6. Results: Migration and malaria

“Gametocyte presence was observed in 42.4% of the cases based on microscopic parasite diagnosis”

A large section of the discussion is devoted to gametocytes and onwards transmission of malaria. However, there is no description on gametocyteaemia in material and methods, and I cannot see any results regarding gametocytaemia on any of the Figures or Tables presented to support the argument. If the authors want to make a statement on gametoctaemia, I request that some results supporting the argument be presented in main tables or provide a bit of description on the methodology section (or describe the gaemtocytaemia methodology as a supplemental file).

7. Table 2:

This table is a screenshot and appears a picture file. Perhaps, the authors can consider creating a table.

8. Table 4:

a. Explain what NI is and what the asterisk after NI is indicating

b. What the column represent? I can see that denominator is 501. It is important to clarify the percentage represents the cases for the given age-group out of the overall cases during the study period.

9. Results: Signs and Symptoms (and Table 6)

How are these signs and symptoms different from what is already known about the disease?

10. Table 6:

Again, I have checked the first two estimates. They are correct and the 95% CI agrees reasonably well. However, “±” notation is confusing. I suggest using 68.1 (63.2 – 73.0) notation to avoid confusion.

Reviewer #2: The manuscript needs to be edited because it includes an excesive number of pages, references and figures.

In the abstract the authors affirm "The objective of this study was to determine the ecoepidemiological

factors associated with the frequency and distribution of malaria transmission in the Darién Region". However, the study results presented in the manuscript discribe very few "eco-epidemiologícal factors" and do not provide results to support the existence of epidemiological or statistical associations. The authors should include addtional results or adjust the objective of the study to a more realistic objective.

Data presented in figure 2 and table 1 correspond to a smaller study that uses a subset of the data but does not contribute new information to the study.

For table 3 and figure 4 It is importante to know the size of the populations at risk used to calculate some of the malaria epidemiological indexes.

The use of 95% confidence intervals in tables 4, 5 and 6 is inadequate and does not contribute usefull information to the data analysis.

Data presented in Table 4 corresponds to the case distribution by age group and should not be used to make inferences about the incidence of malaria by age group without knowing the size of the population at risk in each age group.

The description of symptoms (Table 6) mentions what is already known worldwide about the symptoms of malaria and does not contribute new or usefull information.

The discussion section is very long and difficult to read. Some of the aspects discussed about migration, climate, antimalarial treatment and control measures are not supported with study results.

6. PLOS authors have the option to publish the peer review history of their article (what does this mean?). If published, this will include your full peer review and any attached files.

Reviewer #1: No

Reviewer #2: No

---

## [Author Response · Author response to Decision Letter 0]

6 Sep 2019

Panama, July 8, 2019

Doctor

Luzia Helena Carvalho, PhD.

Academic Editor PLOS ONE

1160 Battery Street

Koshland Building East, Suite 225

San Francisco, CA 94111, USA

Dear Editor

Please find below the corrections made to the manuscript PONE-D-19-14773 “Study of the epidemiological behavior of malaria in the Darién Region, Panama. 2015 - 2017. Epidemiology of malaria in Darién”, following the recommendations and observations of the reviewers and editors of PLOS ONE.

Initial editorial review:

1. The manuscript was corrected according to the style requirements of PLOS ONE.

2. The declaration of ethics was modified according to the recommendations and style of PLOS ONE.

3. The way the informed consent was obtained was explained.

4. The following references were corrected:

• https://doi.org/10.1186/s12936-018-2235-3 (Climatic fluctuations and malaria transmission dynamics, prior to elimination, in Guna Yala, República de Panamá).

• https://doi.org/10.1155/2018/3954717 (Evaluating Malaria Prevalence Using Clinical Diagnosis Compared with Microscopy and Rapid Diagnostic Tests in a Tertiary Healthcare Facility in Rivers State, Nigeria).

• https://doi.org/10.1038/s41598-018-23801-9 (Micro-epidemiology of mixed-species malaria infections in a rural population living in the Colombian Amazon region).

• https://doi.org/10.1155/2018/9163543 (Insecticide Resistance and Its Intensity in Populations of Malaria Vectors in Colombia).

5. Figures 1,2,4 and 6, and all maps were prepared by the Geographic Information Department of ICGES.

6. Figure 7 was eliminated due to the short time to achieve the signature of the participants and to be able to send the manuscript on the date indicated by PLOS ONE.

Reviewer 1.

Major Comments

1. Objective and presentation of results

• The objective was corrected and modify according to the results of the study.

• The results indicated that the intervention measures used by NMP were the use of IRS, application of insecticide through the use of thermospray in places where outbreaks or cases of malaria were registered, application in anopheline breeding grounds of Bacillus sphaericus (Vectolex®) and the active and passive search for suspicious cases.

• The evaluation was carried out taking into consideration the regional malaria control plan, intersectoral cooperation, coverage of interventions, access to malaria diagnosis and treatment and the epidemiological surveillance system.

2. Materials and methods

• The description of the study design was corrected as follows: A retrospective analysis was carried out to determine the incidence, prevalence and distribution of malaria in the Darién region, considering all cases diagnosed by active and passive search by the PNM during the 2015-2017 period.

• In the paragraph that indicates "Malaria is a disease of mandatory notification in Panama", this includes all cases of malaria detected by active search by the NMP and passive by health facilities.

3. Tables 4,5 and 6.

The confidence intervals in tables 4,5 and 6 were eliminated considering the suggestion of Reviewer 2.

Minor Comments

1. Summary

• Information on the age group and sex with malaria, the sentence was corrected as follows: 62.7% (n = 314) of the cases occurred in males and 65.5% (n = 328) of the total cases occurred in economically active age groups.

2. Introduction: paragraph 1-4

• The words autochthonous and Mesoamerica / anthropogenic were defined

• The National Malaria Program (NMP) was placed in parentheses when it was first written.

• Point (.) Was removed on line 3 of paragraph 4.

• The word ethnicity was deleted in order to make it clear that all human groups are ethnic groups.

3. API, SPR and ABER were defined. It was specified how each of these indicators were calculated.

4. Results: Malaria indicators

• In reference to the age group of 1 to 59 years that had a higher incidence of malaria, the sentence was corrected, as follows: People between the ages of 1 to 59 years of age presented a higher malaria incidence (n = 455 ; 91.0%) compared to the rest of the age groups.

5. Results: Migration and malaria

• Regarding observations on gametocytes, in diagnostic and treatment methods, we added that the number of samples with the presence of gametocytes was considered for analysis due to their epidemiological importance. In the results of diagnosis and treatment, we also added that 43.5% (n = 218) of the diagnosed cases showed the presence of gametocytes, and in Table 4, the number of cases with the presence of gametocytes was added.

6. Table 2

• A table was created for the image capture of Table 2, This Table was divided into two tables (Table 1 and Table 1a).

7. Table 4 (now Table 3)

In Table 3; NI = There is no information about the age group. The asterisk (*) was removed.

b. which represents the column.

In the column the title was corrected and “Total cases by age” was placed, where reference is made to the total number of cases by age group.

8. Results: signs and symptoms (and table 6)

It is important to indicate that although the signs and symptoms of malaria are known in general, they can vary and present a broad clinical spectrum that can result in asymptomatic infection, uncomplicated and fatal malaria according to the species of the parasite and its genotype, the human host and the environment. Similarly, people who live in endemic sites and especially adults get to develop some immunity compared to people who live in non-malarial sites where the disease can manifest itself in the population in a more severe way. For these reasons, it was considered to study and analyze the main signs and symptoms of the disease in people diagnosed with malaria. This type of analysis has not been done in this region of the country; thus we consider it important.

9. Table 6

The confidence interval columns in the table were eliminated following recommendation of Reviewer 2.

Reviewer 2.

• The manuscript was edited, and several paragraphs were deleted to shorten it.

• In summary, the objective was corrected as follows: The objective of the study was to determine the main characteristics of the epidemiological behavior of malaria in the Darién region.

• Figure 2 and Table 1 were removed considering the reviewer's suggestion.

• The population size was added in Table 2.

• In Tables 3,4 and 5, the columns with the confidence intervals were eliminated.

• The data in Table 3 was only used to analyze the number of cases by age group and sex. 

•Table 5, about signs and symptoms of malaria: Although the signs and symptoms of the disease are generally known, however, as indicated above, they can vary and present a broad clinical spectrum that can result in asymptomatic infection, uncomplicated and lethal malaria according to the species of the parasite and its genotype, the human host and the environment. Similarly, people who live in endemic sites and especially adults can develop some immunity compared to people who live in non-malarial sites where the disease can manifest itself in the population in a more severe way. For these reasons, it was determined to know and analyze the main signs and symptoms of the disease in people diagnosed with malaria. This type of analysis has not been done in this region of the country, so we consider it important to analyze it to see if with subsequent studies that are done you can find differences.

• The discussion was shortened. Aspects related to migration, antimalarial treatment and control measures were modified according to the results.

Thank you again for the review of the manuscript. We hope that the comments have been appropriately addressed and that the manuscript is now acceptable for publication. 

Yours sincerely,

Lorenzo Cáceres

---

## [Decision Letter · Decision Letter 1]

24 Sep 2019

PONE-D-19-14773R1

Study of the epidemiological behavior of malaria in the Darien Region, Panama. 2015 - 2017. Epidemiology of malaria in Darien

PLOS ONE

Dear Dr Cáceres,

Thank you for submitting your manuscript for review to PLoS ONE. After careful consideration, we feel that your manuscript will likely be suitable for publication if the authors revise it to address specific point raised now by the reviewers.  According to reviewers, there are still some minor areas where further improvements would be of substantial benefit to the readers, including study design and tables. Finally, the manuscript should go through an in-depth proofreading.

We would appreciate receiving your revised manuscript by October 10. To enhance the reproducibility of your results, we recommend that if applicable you deposit your laboratory protocols in protocols.io, where a protocol can be assigned its own identifier (DOI) such that it can be cited independently in the future. For instructions see: http://journals.plos.org/plosone/s/submission-guidelines#loc-laboratory-protocols

We look forward to receiving your revised manuscript.

Kind regards,

Luzia Helena Carvalho, Ph.D.

Academic Editor

PLOS ONE

Reviewers' comments:

Reviewer's Responses to Questions

**Comments to the Author**

1. If the authors have adequately addressed your comments raised in a previous round of review and you feel that this manuscript is now acceptable for publication, you may indicate that here to bypass the “Comments to the Author” section, enter your conflict of interest statement in the “Confidential to Editor” section, and submit your "Accept" recommendation.

Reviewer #1: All comments have been addressed

Reviewer #2: All comments have been addressed

2. Is the manuscript technically sound, and do the data support the conclusions?

Reviewer #1: Yes

Reviewer #2: Yes

3. Has the statistical analysis been performed appropriately and rigorously? 

Reviewer #1: Yes

Reviewer #2: Yes

4. Have the authors made all data underlying the findings in their manuscript fully available?

Reviewer #1: Yes

Reviewer #2: Yes

5. Is the manuscript presented in an intelligible fashion and written in standard English?

Reviewer #1: No

Reviewer #2: Yes

6. Review Comments to the Author

Reviewer #1: Dear authors,

The manuscript is much improved for clarity.

There are still some minor areas where further improvements would be of substantial benefit to the readers.

The following description of the study design appears in abstract and main text: “We conducted a retrospective analysis to determine the incidence, prevalence, and malaria distribution in the Darien region” It appears that the authors have indeed estimated the incidence of the disease as they have divided the number of positive slides in the year by the total population at the start of the year (In Malariometric indicators section).

I request the authors to add further minor clarifications.

1. Strictly, the term incidence is the ratio of the number of new cases of the disease within the time-period divided by the number of people who are initially free of the disease at the start of the year.

It might not have been possible to know exactly how many were free of malaria at the start of the study. The authors should clearly state that they made an assumption that everybody was free of malaria at the start of the study period.

2. The word “prevalence” seem to be present throughout the manuscript. It would be appropriate if the authors can clarify if the word “prevalence” was used loosely or if they had indeed made an attempt to estimate “prevalence”. If the word “prevalence” was used in a loose sense, then I suggest deleting it to avoid confusion to the readers. If the intention was to estimate prevalence, then it should be appropriately explained in the methodology section.

Reviewer #2: Table 1a should be named as continuation of table 1. No need to call it 1a.

Table 2 should have the description for the different malariometric indicators, and figure 4 for API.

Table 4, the colum named "No cases with gametocytes" should be moved to the right of the table, so the % of diagnosed cases and cummulative percent figures make sence.

Table 5, symptoms in cases of vivax and falciparum should not be addedd. They should be described independently and compared. Symtoms not neccesarly are the same in vivas and falciparum. I suggest to add one column and calculate the % distribution of symptoms for ecah type of infection and made the comparison.

7. PLOS authors have the option to publish the peer review history of their article (what does this mean?). If published, this will include your full peer review and any attached files.

Reviewer #1: No

Reviewer #2: Yes: Carlos Rojas, MD, PhD, Professor of Epidemiology

---

## [Author Response · Author response to Decision Letter 1]

11 Oct 2019

Panama, October 8, 2019

Doctor

Luzia Helena Carvalho, PhD.

Academic Editor PLOS ONE

1160 Battery Street

Koshland Building East, Suite 225

San Francisco, CA 94111, USA

Dear Editor

Please find below the corrections made to the manuscript PONE-D-19-14773 “Study of the epidemiological behavior of malaria in the Darién Region, Panama. 2015 - 2017. Epidemiology of malaria in Darién”, following the recommendations and observations of the reviewers and editors of PLOS ONE.

Reviewer #1: 

As suggested, the word “prevalence” was eliminated and replace by the word “incidence” throughout the manuscript. 

Reviewer #2:

Referring to Table 1., the word Table 1a was deleted and added below.

In Table 2, the definitions of the API, SPR and ABER malariometric indicators were added.

In Table 4, the “No. of gametocyte” column was moved to the right side.

In Table 5, a column with the “signs and symptoms” for P. falciparum was added and separated from P. vivax. Also a comparison was made.

---

## [Editor Report · Decision Letter 2]

16 Oct 2019

Study of the epidemiological behavior of malaria in the Darien Region, Panama. 2015 - 2017. Epidemiology of malaria in Darien

PONE-D-19-14773R2

Dear Dr. Cáceres,

We are pleased to inform you that your manuscript has been judged scientifically suitable for publication and will be formally accepted for publication once it complies with all outstanding technical requirements.

With kind regards,

Luzia Helena Carvalho, Ph.D.

Academic Editor

PLOS ONE
---

## [Editor Report · Acceptance letter]

8 Nov 2019

PONE-D-19-14773R2 

Study of the epidemiological behavior of malaria in the Darien Region, Panama. 2015 - 2017. 

Dear Dr. Cáceres Carrera:

I am pleased to inform you that your manuscript has been deemed suitable for publication in PLOS ONE. Congratulations! Your manuscript is now with our production department. 

With kind regards,

on behalf of

Dr. Luzia Helena Carvalho 

Academic Editor

PLOS ONE